# A New Synthesized Dicarboxylated Oxy-Heparin Efficiently Attenuates Tumor Growth and Metastasis

**DOI:** 10.3390/cells13030211

**Published:** 2024-01-23

**Authors:** Li Li, Uri Barash, Neta Ilan, Malik Farhoud, Xiao Zhang, Israel Vlodavsky, Jin-Ping Li

**Affiliations:** 1Shenzhen Hepalink Pharmaceutical Group Co., Ltd., Shenzhen 518057, China; lili@hepalink.com; 2Technion Integrated Cancer Center, Rappaport Faculty of Medicine, Technion, Haifa 3525422, Israel; ubarash@yahoo.com (U.B.); netailan@technion.ac.il (N.I.); malik.far.96@gmail.com (M.F.); 3Department of Medical Cell Biology, Uppsala University, 751 23 Uppsala, Sweden; xiao.zhang@uu.se; 4Department of Medical Biochemistry and Microbiology, SciLifeLab, Uppsala University, 751 23 Uppsala, Sweden

**Keywords:** heparanase, primary tumor growth, metastasis, tumor growth inhibition, mesothelioma, breast carcinoma, pancreatic carcinoma, chemically modified heparin

## Abstract

Heparanase (Hpa1) is expressed by tumor cells and cells of the tumor microenvironment and functions to remodel the extracellular matrix (ECM) and regulate the bioavailability of ECM-bound factors that support tumor growth. Heparanase expression is upregulated in human carcinomas, sarcomas, and hematological malignancies, correlating with increased tumor metastasis, vascular density, and shorter postoperative survival of cancer patients, and encouraging the development of heparanase inhibitors as anti-cancer drugs. Among these are heparin/HS mimetics, the only heparanase-inhibiting compounds that are being evaluated in clinical trials. We have synthesized dicarboxylated oxy-heparins (DCoxHs) containing three carboxylate groups per split residue (DC-Hep). The resulting lead compound (termed XII) was upscaled, characterized, and examined for its effectiveness in tumor models. Potent anti-tumorigenic effects were obtained in models of pancreatic carcinoma, breast cancer, mesothelioma, and myeloma, yielding tumor growth inhibition (TGI) values ranging from 21 to 70% and extending the survival time of the mice. Of particular significance was the inhibition of spontaneous metastasis in an orthotopic model of breast carcinoma following resection of the primary tumor. It appears that apart from inhibition of heparanase enzymatic activity, compound XII reduces the levels of heparanase protein and inhibits its cellular uptake and activation. Heparanase-dependent and -independent effects of XII are being investigated. Collectively, our pre-clinical studies with compound XII strongly justify its examination in cancer patients.

## 1. Introduction

Heparanase is an endoglucuronidase that cleaves heparan sulfate (HS), thereby altering the structure and function of heparan sulfate proteoglycans (HSPG) and contributing to tumor-mediated remodeling of the cell surface and extracellular matrix (ECM) [1,2,3]. These actions dynamically impact multiple pro-tumorigenic regulatory pathways, most notably by augmenting the bioavailability of growth factors and cytokines bound to heparan sulfate in the tumor microenvironment [2,4,5,6,7,8]. In addition, heparanase upregulates the expression of multiple genes (i.e., VEGF, HGF, RANKL, MMP-9, IL-2, and IFN-γ) that promote aggressive tumor progression and inflammation [9]. Intense research efforts during the last two decades revealed that heparanase expression is upregulated in various human carcinomas, sarcomas, and hematological malignancies [4,5,6,8,10]. Compelling evidence ties heparanase to all steps of tumor formation, including tumor initiation, growth, metastasis, and chemoresistance [11,12,13]. Likewise, many clinical association studies have consistently demonstrated that upregulation of heparanase expression correlates with increased tumor size, tumor angiogenesis, enhanced metastasis, and poor prognosis [14,15,16,17], providing strong support for the pro-tumorigenic function of the enzyme. By contrast, knockdown of heparanase or treatments of tumor-bearing mice with heparanase-inhibiting compounds markedly attenuate tumor progression, underscoring the potential of anti-heparanase therapy for multiple types of cancer and encouraging the development of heparanase inhibitors as anti-cancer drugs [12,14]. Heparin/HS-mimetics (i.e., Muparfostat = PI-88, Roneparstat = SST0001, Necuparanib = M402, Pixatimod = PG545) that inhibit heparanase enzymatic activity were and are being evaluated in clinical trials for various types of cancer and appear to be well tolerated [18,19,20,21]. Heparanase-inhibiting small molecules [22] and heparanase-neutralizing monoclonal antibodies were developed, in part based on the crystal structure of the heparanase protein [23], and are being examined in pre-clinical studies.

Heparin, exclusively produced by mast cells, is a highly sulfated form of HS. Early studies evidenced unfractionated heparin (UFH) as a substrate and potent heparanase inhibitor [24]. In subsequent studies, investigations were mainly oriented to generate non-anticoagulant heparin derivatives to overcome the anticoagulation and bleeding side effects and to improve the pharmacokinetics and bioavailability of newly generated species of heparin. Among these compounds is Roneparstat (=SST0001), a chemically modified 100% N-desulphated, N-reacetylated, and 25% glycol-split heparin with very low anticoagulant activity and a molecular weight ranging from 15,000 to 25,000 Da [25]. Roneparstat was highly effective in murine models of myeloma [12]. Based on this preclinical evidence, a multicenter, phase I study was performed to assess the safety and tolerability profile of Roneparstat in patients with relapsed/refractory myeloma (clinicaltrials.gov identifier: NCT01764880). The compound exhibited excellent safety and tolerability profiles and could be administered at dose levels of 200 and 400 mg/patient/day without causing any clinically relevant toxicity [18]. It was concluded that even though the phase I study did not provide evidence of a direct anti-myeloma effect of Ronepartstat in humans, exploration of Roneparstat in combination regimens for the treatment of myeloma, and possibly other types of cancer, was justified [21]. Of note, a recent phase 1b clinical trial (PG545 = Pixatimod in combination with nivolumab = anti-PD-1 drug) showed signs of clinical benefit for colorectal patients [19].

To increase the affinity of glycol-split residues for the active site of heparanase and improve the pharmacokinetic properties of Roneparstat, dicarboxylated oxy-heparins (DCoxHs) containing three carboxylate groups per split residue (DC-Hep) were prepared [25]. The resulting lead compound (XII) was selected for upscaling, characterization, and examination in mouse tumor models toward further development and investigation of its mode of action. Promising results were obtained in models of pancreatic carcinoma, breast cancer, mesothelioma, and myeloma as a single agent, yielding tumor growth inhibition (TGI) values ranging from 21% to 70%. Of particular significance was the inhibition of spontaneous metastasis in an orthotopic model of breast carcinoma following resection of the primary tumor. It appears that apart from inhibition of heparanase expression levels and enzymatic activity, compound XII competes with HS on the cell surface and thereby inhibits cellular binding, uptake, and processing/activation of the secreted 65 kDa latent form of the enzyme. Collectively, our pre-clinical studies with compound XII justify progress into phase I and hopefully more advanced clinical trials in cancer patients.

## 2. Materials and Methods

### 2.1. Preparation of Compound XII

Using sodium heparin as the starting material, the sulfonic acid group at position 2 of uronic acid was removed under alkaline heating conditions to form a 2,3 epoxy structure. The epoxy was then hydrolyzed under neutral conditions by heating to generate a vicinal diol structure followed by oxidation to break the C–C bond of the ortho-diol structure to generate a derivative with two aldehyde groups. Finally an oxidizing agent was added to oxidize the aldehyde group into a carboxyl group to obtain the heparin derivative containing the characteristic structure of tricarboxylic acid. The products were separated through column chromatography and the components of approximately 10 kDa were collected and purified through ethanol precipitation to obtain compound XII.

### 2.2. Heparanase Enzymatic Activity

#### 2.2.1. Fondaparinux (Arixtra)

The assay colorimetrically measures the appearance of the disaccharide product of heparanase-catalyzed cleavage of the pentasaccharide fondaparinux (Arixtra) using the tetrazolium salt WST-1. Because this assay has a homogeneous substrate with a single point of cleavage, the kinetics of the enzyme can be reliably characterized [26]. Assay solutions (100 μL) were composed of 40 mM sodium acetate buffer (pH 5.0) and 100 mM fondaparinux (Arixtra) with or without inhibitor [26]. Recombinant active heparanase, produced as described in [24], was added to a final concentration of 150 ng/mL to start the assay. The plates were incubated at 37 °C for 4–24 h and the reaction was stopped by the addition of 100 μL solution containing 1.69 mM 4-[3-(4-iodophenyl)-2-(4-nitrophenyl)-2H-5-tetrazolio]-1,3-benzene disulfonate (WST-1) in 0.1 M NaOH. The plates were developed at 60 °C for 60 min, and the absorbance (OD) was measured at 584 nm [26].

#### 2.2.2. FRET Analysis

Heparanase enzymatic activity was determined using homogeneous time-resolved fluorescence energy transfer-based assay (TR-FRET), essentially as instructed by the manufacturer (Cisbio Bioassays S.A.S., Codolet, France) [27]. Recombinant active heparanase (GS3-linker 58 kDa) was purified and used for the assay. Test compounds were dissolved and diluted using buffer containing 50 mM Tris-HCl (pH 7.4), 0.15 M NaCl, 0.1% protease-free BSA, and 0.1% CHAPS. The diluted compound XII (3 µL) was mixed with 3 µL heparanase (0.9 ng) in 0.2 M acetic acid buffer (pH 5.5) and 6 µL of assay buffer (pH 5.5) [27] to a final volume of 12 µL in a 384-well low-volume microplate. The enzyme activity was detected by determining the HTRF signal ratio of 665/620 nm in a FLUOstar OMEGA reader (BMG Labtech, Ortenberg, Germany). Each reaction was run in triplicate.

#### 2.2.3. ^35^Sulfate-Labeled ECM

Extracellular matrix (ECM) substrate deposited by cultured corneal endothelial cells *closely* resembles the subendothelial basement membrane in its composition, biological function, and barrier properties. Briefly, metabolically sulfate (^35^S)-labeled ECM deposited by corneal endothelial cells was incubated with recombinant heparanase or cell lysate, and HS degradation fragments released into the incubation medium were analyzed by gel filtration. Detailed information about the preparation of sulfate-labeled ECM and its use for the heparanase assay has been previously described [28,29,30,31]. The assay is highly sensitive and suitable for measurements of heparanase activity in intact cells, extracts of cells and tissues, plasma, and bodily fluids. It is semi-quantitative and time-consuming. Briefly, sulfate [^35^S]-labeled ECM coated on the surface of 35 mm tissue culture dishes was incubated (4 h, 37 °C, pH 6.0, 1 mL final volume) with recombinant active heparanase [24] (2.0 ng/mL) in the absence and presence of the inhibitory molecule. The reaction mixture contained 50 mM NaCl, 1 mM DTT, 1 mM CaCl_2_, and 10 mM phosphate-citrate buffer, pH 6.0. To evaluate the occurrence of proteoglycan degradation, the incubation medium was collected and applied for gel filtration on a Sepharose 6B column (0.9 × 30 cm). Fractions (0.2 mL) were eluted with PBS and counted for radioactivity. The excluded volume (Vo) was marked by blue dextran and the total included volume (Vt) was marked by phenol red. Degradation fragments of HS side chains were eluted from Sepharose 6B at 0.5 < Kav < 0.8 [29].

### 2.3. Cells

Human (Mia PaCa-2, BxPC-3, AsPC-1) and mouse (Panc02) pancreatic cancer cell lines, human (MDA-MB-231) and mouse (4T1, Luc4T1) breast carcinoma cell lines, the human MSTO-211H mesothelioma cell line, the human SiHa cervical carcinoma cell line, and the mouse MPC-11 myeloma/plasmacytoma cell line were purchased from the American Type Culture Collection. Cell lines were confirmed to be pathogen-free, and human cell lines were authenticated to confirm their origin before use. Cells were cultured in Dulbecco’s Modified Eagle Medium (DMEM) or RPMI medium (Biological Industries, Beit Haemek, Israel) containing 10% FBS and maintained at 37 °C in a humidified incubator with 5% CO_2_ and 95% air. Cells were passed in culture no more than 2 months after being thawed from authentic stocks. Preparation of cell/tissue lysates and immunoblotting were performed essentially as described.

### 2.4. Cancer Models

Mice were housed in a pathogen-free facility with access to food and water ad libitum. All animal studies were performed in compliance with the regulations and ethical guidelines for experimental animal studies, in accordance with the Technion’s Institutional Animal Care and Use Committee (IL-078-05-21; OPRR-A5026-01). At the time of sacrifice, tumors were removed to determine the primary tumor burden. Gross metastatic events were identified through visual inspection of lymph node basins, liver, peritoneum, diaphragm, lungs, and spleen. Samples were fixed in 10% formalin or snap-frozen in liquid nitrogen for further studies.

#### 2.4.1. Pancreatic Ductal Adenocarcinoma (PDAC)

Human (Mia PaCa-2, BxPC-3, AsPC-1, incompetent model) PDAC cells from exponential cultures were dissociated with trypsin/EDTA, washed with PBS, and brought to a concentration of 5 × 10^7^ cells/mL. The cell suspensions (5 × 10^6^/0.1 mL) were subcutaneously inoculated in the right flank of 6–8-week-old SCID mice, respectively. When palpable tumors were noticed (Days 8–10), mice were randomly assigned to cohorts (8 mice each) receiving vehicle, compound XII (20–160 mg/kg/once a day, s.c.), or gemcitabine (30 mg/kg twice weekly or 50 mg/kg weekly). Tumor size was determined twice or thrice a week by externally measuring tumors in two dimensions using a caliper. Tumor volume (V) was determined by the equation V = L × W^2^ × 0.52, where L is the length and W the width of the xenograft. At the end of the experiment, mice were sacrificed and xenografts were resected, weighed, and fixed in formalin for pathological examination and immunostaining.

#### 2.4.2. PDX Model

Tumor fragments from donor mice were harvested and used for inoculation into mice, essentially as described in [32]. Each mouse was subcutaneously inoculated in the right upper flank with primary human pancreatic PDX PA1233 tumor fragment (2–3 mm in diameter) [32]. Before dosing with drugs, all animals were weighed and tumor volumes were measured with calipers. When the average tumor volume reached 128.0 mm^3^, the mice were randomized into groups according to their tumor sizes. Since the tumor volume would affect the effectiveness of the treatment, randomization was based on tumor volume to ensure that the tumor volumes between different groups were similar.

#### 2.4.3. Breast Carcinoma

Luc-4T1 murine breast carcinoma cells were injected (5 × 10^5^ cells/mouse) into the third mammary fat pad of BALB/c mice. Two days after cell inoculation, mice were divided into groups (*n* = 8 mice/group) and treated (i.p.) with vehicle (PBS), Docetaxel (10 mg/kg, i.p. QWx4), or compound XII (i.p. 20–80 mg/kg/once a day). The primary tumors were excised on Day 13 and weighed. Treatment continued as described above and the mice were evaluated for the formation of lung metastases (Day 28—IVIS bioluminescent imaging) and survival time. MDA-MB-231 human breast carcinoma cells (5 × 10^6^/mouse, *n* = 8–10/group) were inoculated orthotopically into SCIS/NOD mice. Mastectomy was performed on Day 13 and the mice were monitored for tumor volume and survival time.

#### 2.4.4. Mesothelioma

Luciferase-labeled MSTO-211H human mesothelioma cells were i.p. inoculated (2 × 10^6^/0.2 mL) into NOD/SCID mice. Ten days after cell inoculation, mice were randomly assigned to cohorts (7 mice each) receiving vehicle, compound XII (s.c. 80 mg/kg/once a day), or cisplatin (3 mg/kg; once/3 weeks), all administered i.p. Tumor development was inspected (once a week) by IVIS imaging following administration of luciferin [14].

#### 2.4.5. Myeloma

MPC-11 mouse myeloma/plasmacytoma cells [33] were injected (0.5 × 10^6^/mouse, s.c., *n* = 8/group) into BALB/c mice. Mice were treated with compound XII (40 mg/kg; once a day, starting on Day 3). Tumor volume was measured on Days 9, 11, and 14. Tumors were excised on Day 14 and weighed.

#### 2.4.6. IVIS Imaging

Bioluminescent imaging of luciferase-expressing tumors was performed using a highly sensitive, cooled charge-coupled device (CCD) camera mounted in a light-tight specimen box (IVIS; Xenogen Corp., Waltham, MA, USA). Imaging was performed in real-time, was non-invasive, and provided quantitative data. Briefly, mice were intraperitoneally injected with D-luciferin substrate at 150 mg/kg, anesthetized, and placed onto a warmed stage inside the light-tight camera box, with continuous exposure to isoflurane (EZAnesthesia, Palmer, PA, USA). Light emitted from bioluminescent cells was detected by the IVIS camera system with images quantified for tumor burden using a log-scale color range set at 5 × 10^4^ to 1 × 10^7^ and total photon counts per second (PPS) was measured using Living Image (Xenogen) [14].

### 2.5. Western Blot Analysis

Nude mice (*n* = 6/group) s.c. inoculated with Mia PaCa-2 cells were treated with increasing amounts of compound XII (daily injections of 20, 40, or 80 mg/kg) or vehicle alone. On Day 25, the tumors were resected and the tumor tissues (50 mg) were lysed in 500 µL of RIPA buffer. After incubation on ice for 30 min, the lysates were centrifuged and supernatants were collected. Total protein was determined using the BCA method and 50 µg of protein was applied to SDS-PAGE. Western blot analysis was carried out using anti-heparanase antibody (Proteintech 66226-1-ig, Rosemont, IL, USA) and anti-GAPDH polyclonal antibodies.

### 2.6. Heparanase Uptake and Processing

Uptake experiments were carried out essentially as described in [30]. Briefly, cells (SiHa cervical carcinoma) were incubated with 65 kDa latent heparanase purified as described in [28] (1 μg/mL; serum-free conditions) in the absence and presence of increasing concentrations (1, 5, 10 ug/mL) of compound XII. Twenty-four hours later, the medium was aspirated, cells were washed three times with ice-cold PBS, and cell extracts were subjected to immunoblotting with polyclonal anti-heparanase antibody (ANT-155, ProSpec-Tany, TechnoGene Ltd., Ness-Ziona, Israel), which detects both the latent (65 kDa) and active (8 + 50 kDa) enzyme.

### 2.7. Pharmacokinetics

A total of 24 Sprague-Dawley rats (12 females, 12 males) were subcutaneously or intravenously injected with a single dose of compound XII. For the intravenous injection and subcutaneous injection, each animal was given 15 mg/kg. This dosage was based on the calculation of surface area between mouse and rat. Selecting the lowest effective dose (20 mg/kg in mice) was for the consideration of safety when the compound was i.v. injected. Whole blood was collected at 15 min, 30 min, 1 h, 2 h, 4 h, 6 h, 8 h, 12 h, and 24 h post-injection. The concentration in the plasma was measured by LCMS. The plasma sample was processed by liquid–liquid extraction in phenol: methylene chloride (*v*/*v*, 2:1) and then 300 μL of supernatant was transferred to 0.6 mL centrifuge tubes in which 2.00 μL of formic acid was added. Then, the samples were hydrolyzed in a water bath (90 °C) for at least 20 hr. A Waters Xevo TQ-S mass spectrometer with the electrospray ionization source operated in negative scan mode was used to monitor the precursor to product ion transitions of *m/z* 390.03 and 261.60 for compound XII and 5-Fluorouridine (the internal standard), respectively. Phoenix^TM^ WinNonlin 8.1 was used to calculate the PK parameters of compound XII by employing non-compartmental model analysis.

The radioactive ^89^Zr label was attached to the terminal aldehyde group of compound XII. The chemical purity and RCP (radio chemical purity) of ^89^Zr-XII was ≥90%, measured using a 1500 series HPLC coupled with an online radiation detector. The relative biological activity (heparanase-inhibiting activity) of ^89^Zr-XII was equivalent to that of unlabeled compound XII. The tissue distribution profile of ^89^Zr-XII following a single subcutaneous administration in 4T1 tumor-bearing mice was scanned by static PET/CT for 10–30 min at each time point and the images were collected for analysis using the software of PMOD (PMOD Technologies, Billerica, MA, USA).

### 2.8. Statistics

Data are presented as the mean ± SEM. Statistical significance was analyzed by 2-tailed Student’s *t*-test or one-way ANOVA. A value of *p* ≤ 0.05 was considered significant. Datasets passed the D’Agostino–Pearson normality test (Prism 5 utility software, GraphPad, USA).

## 3. Results

### 3.1. Preparation of Dicarboxylated Oxy-Heparin (DcoxyH = compound XII)

Compound XII was prepared by two-step oxidation of heparin. In the first oxidation step, periodate oxidation of heparin’s nonsulfated uronic acid residues led to oxy-heparins (oxyHs), characterized by the split of the C2–C3 linkage and the formation of two aldehyde groups [25]. These were further oxidized to carboxy groups yielding dicarboxylated oxy-heparins (DCoxHs). Oxidation of the aldehyde to carboxyl was performed using sodium chlorite (NaClO_2_) in aqueous media, pH 4, at 0 °C for 24 h [25] (Figure 1).

Compound XII was examined for its capacity to inhibit heparanase enzymatic activity by applying the Arixtra colorimetric assay (Figure 1A), FRET assay (Figure 1B), and sulfate-labeled ECM degradation assay (Figure 1C). For this purpose, recombinant heparanase was incubated with fondaparinux (Arixtra) in the absence and presence of increasing concentrations of compound XII. Cleavage of the pentasaccharide substrate into a disaccharide and trisaccharide products was colorimetrically evaluated at 584 nm following the addition of WST-1 [26]. As demonstrated in Figure 1A, compound XII was highly active in inhibiting the enzyme, yielding an IC50 of 71 ng/mL. Measurements of heparanase enzymatic activity at various concentrations of compound XII using the FRET assay [27] yielded an IC50 of 241 ng/mL (Figure 1B). Compound XII was then examined for inhibition of HS degradation using naturally produced ECM as the substrate. The results are presented as the actual gel filtration profile of sulfate-labeled HS degradation products released into the incubation medium (Figure 1C). Notably, compounds XII and SST0001 (= Roneparstat, a lead anti-heparanase non-anticoagulant glycol-split heparin) [25,34] exhibited comparable heparanase-inhibiting activity. We relied on the ECM assay given that the Arixtra assay uses a synthetic pentasaccharide substrate, while ECM is a naturally produced substrate and hence better predicts the inhibitory effects of compounds in vivo.

### 3.2. Pancreatic Cancer

Heparanase is known to participate in the progression of PDAC, and elevated levels of heparanase in patients with PDAC correlate with worse overall survival [35]. Taking into account that heparanase is contributed by both tumor cells and host immune cells, we applied mouse (Panc02) and human (MiaPaca-2, BxPC-3, AsPC-1) PDAC cells to examine the effect of compound XII in syngeneic (intact host immune system; C57BL/6) and non-syngeneic (impaired immune system; SCID/NOD) backgrounds, respectively.

Several pancreatic cancer models were examined. Depending on the pancreatic cell line, treatment with compound XII (daily 20–160 mg/kg) alone yielded tumor growth inhibition (TGI) ranging from 21% (Mia PaCa-2, 20 mg/kg, QD) to 67% (Mia PaCa-2, 160mg/kg, QD). Intermediate TGIs (about 50%) were obtained with the BxPC-3 PDAC cell line. In related experiments, compound XII (daily 80 mg/kg) yielded 50% prolongation of the median survival of mice inoculated orthotopically with Panc02 cells. A similar 46% median survival prolongation vs. control untreated mice was observed in mice inoculated (i.p.) with AsPC-1 cells and treated with compound XII (daily 100 mg/kg).

In human Mia PaCa-2 tumor-bearing mice, a significant 21% inhibition of tumor growth was noted already at 20 mg/kg of compound XII, increasing to a comparable 45% inhibition when the drug was administered at 40 or 80 mg/kg, and increasing to a comparable 70% inhibition when the drug was administered at 160 mg/kg (Figure 2A). In this setting, compound XII (20 mg/kg) was as effective as gemcitabine (50 mg/kg, *p* < 0.01 vs. control, *p* > 0.05 vs. Gem), while treatment with 40 mg/kg of compound XII was more effective (Figure 2A). In the BxPC-3 model, treatment with compound XII (50 mg/kg) or Gem (30 mg/kg) reduced primary tumor growth by ~50% (each, *p* < 0.01 vs. control) (Figure 2B).

Next, we investigated the therapeutic efficacy of compound XII in the PA1233 human pancreatic cancer PDX model. Briefly, tumor fragments from donor mice were harvested and used for inoculation into female BALB/c nude mice. Each mouse was subcutaneously inoculated in the right upper flank with primary human pancreatic PDX PA1233 tumor fragment (2–3 mm in diameter). On Day 54 after drug administration, the mean tumor volume of mice in the untreated control group was 1543.26 mm^3^. The mean tumor volume of mice treated with compound XII (80 mg/kg) alone was 981.17 mm^3^ (TGI = 39.68%; *p* < 0.01) (Figure 2C). These results indicated that compound XII was well tolerated in the subcutaneous human pancreatic cancer PA1233 PDX model. Our results demonstrated the effectiveness of compound XII in attenuating PDAC progression as a single drug in both immunocompetent and incompetent model systems (Figure 2).

### 3.3. Breast Cancer

Given the involvement of heparanase in the pathogenesis of breast cancer [36,37,38,39,40], we investigated the effect of compound XII on breast tumorigenicity. Briefly, MDA-MB-231 human breast carcinoma cells were inoculated (s.c.) into SCIS/NOD mice. Mice were treated with compound XII starting on Day 10 and the tumor volume was measured twice a week. As demonstrated in Figure 3A, significant inhibition (~20%) of MDA-MB-231 primary tumor growth was noted already at 20 mg/kg of compound XII, increasing to a comparable 58% inhibition when the drug was administered at 40 or 80 mg/kg. All of the animal treatments were well tolerated. Notably, treatment with compound PG545 (= Pixatimod, a cholestanol-sulfo-tetrasaccharide) [41], yielded a TGI of 59-77% on MDA-MB-231 primary tumor growth, but this was associated with 4–11% loss in body weight.

Next, we examined the efficacy of compound XII in a survival model of mouse 4T1 breast cancer. Briefly, 4T1 breast cancer cells were orthotopically inoculated into the third mammary gland of BALB/c mice. Mice were then randomized into 5 groups (*n* = 8) treated with vehicle alone, compound XII (daily injections of 20, 40, or 80 mg/kg starting on Day 1), or docetaxel (weekly, 10 mg/kg, starting on Day 10). All mice were subjected to mastectomy on Day 13. In this model, compound XII-treated animals showed prolonged median survival compared with vehicle control (44-49 vs. 38.5 days) or Docetaxel therapy (42.5 days, *p* = 0.012) (Figure 3B).

### 3.4. Breast Cancer Metastasis

We investigated the effect of compound XII in the 4T1 breast carcinoma model. Briefly, Luc-4T1 murine breast carcinoma cells were injected (50,000 cells/mouse) into the third mammary fat pad of BALB/c mice. Two days after cell inoculation, mice were divided into 3 groups (*n* = 10 mice/group) and treated (i.p.) with vehicle (PBS), Paclitaxel (PTX, 10 mg/kg, once on Day 9), or compound XII (80 mg/kg; once a day). The primary tumors were excised on Day 14 and weighed. A significant ~30% (*p* = 0.009) reduction in primary tumor weight was noted in mice treated with compound XII while the inhibitory effect of PTX was not significant. Treatment continued after removal of the primary tumors and the mice were subjected to IVIS analysis on Day 28. The results (Figure 3C) revealed nearly complete inhibition of metastasis in response to treatment with compound XII alone. Notably, compound XII alone was more effective than PTX. Altogether, while compound XII had only a limited effect on the growth of the primary breast tumors, it was very effective in attenuating the resulting spontaneous metastases.

### 3.5. Mesothelioma

Mesothelioma tumors express high levels of heparanase and exhibit high sensitivity to treatment with heparanase-inhibiting compounds [14], providing a strong rationale for exploring the effect of compound XII on mesothelioma progression. We examined the effect of compound XII or cisplatin (cis) on MSTO-211H human mesothelioma tumor progression. Briefly, luciferase-labeled MSTO-211H human mesothelioma cells (2 × 10^6^) were intraperitoneally (i.p.) inoculated into NOD/SCID mice. Two days after cell inoculation, mice were divided into 3 groups (*n* = 6–7/group) and treated (i.p.) with vehicle (PBS), compound XII (80 mg/kg; once a day), or cisplatin (3 mg/kg; once every 3 weeks). Tumor development was inspected by IVIS once a week, starting on Day 10 post-cell inoculation (when the luciferase signal is detected). Treatment with compound XII or cisplatin alone yielded a comparable 60–80% inhibition of tumor growth (IVIS signal) measured on Days 10, 23, 30, and 36 of the experiment. The actual IVIS signals obtained on Days 10 and 23 of the experiment are presented in Figure 4A and quantified in the bar graphs, revealing highly significant inhibition of tumor growth by XII at all time points. Cisplatin showed no effect on Day 10 but was as effective as compound XII at later time points. Next, we analyzed the effect of the above treatments on the survival of the mice. As demonstrated in Figure 4B (Kaplan–Meier curves), the median survival time of mice treated with compound XII alone was longer than that of control untreated mice (65 vs. 48 days, *p* = 0.0264). Significantly improved survival was also noted in mice treated with cisplatin alone (median survival = 71 days vs. 48 days for untreated mice, *p* = 0.017).

### 3.6. Mouse Myeloma

We examined the effect of compound XII on mouse myeloma tumor growth. Briefly, MPC-11 mouse myeloma/plasmacytoma cells were injected into BALB/c mice. The mice were treated with compound XII and tumor volume was measured on Days 9, 11, and 14 (Figure 5A). Tumors were excised on Day 14 and weighed (Figure 5B). A most pronounced inhibition of tumor growth was observed. All of the animal treatments were well tolerated. The advantage of using a syngeneic mouse model vs. immunodeficient mice is the existence of an intact, fully active immune system that plays an important role in tumor progression and likely also in the anti-tumorigenic effect of heparanase-inhibiting compounds.

### 3.7. Heparanase Levels in PDAC Tumors

As demonstrated in Figure 1, compound XII was a potent inhibitor of heparanase enzymatic activity. Applying the Mia PaCa-2 tumor model, we investigated whether compound XII also affected the levels of heparanase protein in PDAC tumors. For this purpose, mice (n = 6/group) inoculated (s.c.) with Mia PaCa-2 cells were treated with increasing amounts of compound XII (daily injections of 20, 40, or 80 mg/kg) or vehicle alone, as described in Figure 2A. On Day 25, the tumors were resected and the tumor tissue lysates were subjected to Western blotting, applying anti-heparanase and anti-GAPDH polyclonal antibodies. As demonstrated in Figure 6A, treatment with compound XII (20 mg/kg/day) resulted in a marked 50% decrease in the normalized (GAPDH) amount of heparanase. There was no further reduction in response to treatment with higher amounts of compound XII, similar to the corresponding effect on tumor growth presented in Figure 2A.

### 3.8. Heparanase Uptake and Processing

We previously demonstrated that secreted or exogenously added latent 65 kDa heparanase rapidly interacts with heparan sulfate (HS) on the cell surface, followed by internalization and processing into a highly active 50 + 8 kDa enzyme, a process that is used by cells to attenuate the extracellular accumulation of heparanase [30]. We examined the ability of compound XII to attenuate the uptake and processing of exogenously added recombinant latent heparanase. For this purpose, cervical carcinoma cells (SiHa) were left untreated (Con) or were incubated with latent 65 kDa heparanase (1 ug/mL) together with increasing concentrations (1, 5, 10 ug/mL) of compound XII. After 24 h, cultures were washed, and cell lysate samples were subjected to immunoblotting, applying anti-heparanase (upper panel) and anti-actin (lower panel) antibodies. As demonstrated in Figure 6B, complete inhibition of heparanase cellular uptake and processing, reflected by the 50 kDa heparanase subunit, was obtained in the presence of 10 ug/mL of compound XII. Shorter incubation times (i.e., 30–60 min) were needed to detect both the unprocessed 65 kDa and processed 50 kDa forms of the internalized enzyme [30].

### 3.9. Effect of Compound XII on Coagulation Parameters

Compound XII had very low anticoagulant activity, with about 20 USP unit/mg anti-IIa activity and no anti-Xa activity. To evaluate the effect in vivo, 6 male and 6 female Sprague-Dawley rats were subjected to daily repeat-dose subcutaneous injections for 4 weeks. Measurement of coagulation parameters on Day 29 (after last injection) showed a dose-dependent prolongation of aPTT (Figure 7), although the animals did not show obvious bleeding. On Day 57, the aPTT value essentially returned to normal, indicating no accumulation of the compound. The level of fibronectin (FIB) was marginally increased with increased dosage, which may have been a protective feedback for the increased aPTT. Regardless, the level returned to normal after the end of treatment. Prothrombin time (PT) was not affected. The same behavior was observed with male (Figure 7, left) and female (Figure 7, right) rats.

### 3.10. Pharmacokinetics of Compound XII

For evaluation of pharmacokinetics, the compound was injected (s.c. or i.v.) into Sprague-Dawley rats, as described in ‘Materials & Methods’. Whole blood was collected at the time points indicated in Figure 8A. The plasma concentration was measured by LCMS. After a single intravenous injection of 15 mg/kg of compound XII, the mean value of drug exposure (AUClast) of compound XII in Sprague-Dawley rats was 56,951 hr*ng/mL, with a half-life (t1/2) of 0.497 hr and clearance (CL) of 4.42 mL/min/kg, which was about 12.0% of renal blood flow (renal blood flow for rats is 36.8 mL/min/kg [42]). The apparent volume of distribution (Vss) was 0.138 L/kg, which was about 0.207 times the rat total body fluid volume (rat total body fluid volume is 0.668 L/kg [42]). After a single subcutaneous administration of 15 mg/kg of compound XII, the AUClast in plasma was 75,695 hr*ng/mL, with a t1/2 of 1.48 h and bioavailability of 139%. The Tmax was 1 h and Cmax was 30,067 ng/mL. The PK parameters showed no significant difference between gender (Figure 8A).

To examine the distribution, ^89^Zr-labeled compound XII was injected into mice bearing 4T1 breast cancer tumors. Following single subcutaneous administration, the animals were subjected to PET/CT scans at the indicated time points (Figure 8B). Determination of tradioactivity showed distinct distribution of the compound in organs, mainly in the kidney, bone/joint, and liver. The radioactivity peak in the bladder at 2 h post-injection indicated that the compound was rapidly filtered through the kidney. The radioactivity was gradually reduced, but it still accumulated in the kidney up to 72 h. Notably, the radioactivity was elevated with time in the bone and joint up to 72 h post-injection. Nevertheless, radioactivity was constantly detected in the tumor tissue even 72 h post-injection, albeit at a relatively lower level.

## 4. Discussion

For the most highly aggressive human cancers, such as mesothelioma, glioma, and pancreatic adenocarcinoma, durable remission is difficult to achieve and cure remains elusive. This underscores the need to identify new therapeutic targets and strategies for hitting those targets.

Immunohistochemistry, in situ hybridization, PCR, and Western blot analysis have demonstrated that heparanase expression is enhanced in almost all cancers examined to date, including, for example, myeloma, mesothelioma, breast, ovarian, pancreatic, gastric, colon, bladder, brain, prostate, and liver cancers [5,6]. Numerous clinical association studies have consistently demonstrated that upregulated heparanase expression correlates with increased tumor size, tumor progression, enhanced metastasis, and poor prognosis [5,6]. Notably, there is only a single, enzymatically active form of heparanase in humans, it is expressed in very low levels in most normal tissues, and heparanase knock-out animals exhibit no obvious deficits [43]. These characteristics imply that inhibition of heparanase will cause minimal side effects in cancer patients. It should be admitted, however, that anti-heparanase-based therapy has not yet been implemented in a clinical setting. Notably, all anti-heparanase compounds that were and are being examined in clinical trials are heparin-like saccharides. While these compounds were very successful in numerous mouse models, side effects (i.e., bleeding complications, body weight loss) and poor pharmacokinetics halted their clinical advancement in humans. Development of more specific oligosaccharides [44], small molecules [22] and neutralizing monoclonal antibodies is ongoing. Roneparstat (= SST0001), a chemically modified 100% N-desulphated, N-reacetylated, and 25% glycol-split heparin, showed a promising anti-cancerous effect in murine models of myeloma and sarcoma, either alone or in combination with other drugs [25], yet despite being well tolerated, there was insignificant benefit in phase I clinical trials [18].

The current study focuses on developing a new generation of heparin derivatives to increase the affinity to the heparanase protein. For this purpose, dicarboxylated oxy-heparins (DCoxHs) containing three carboxylate groups per split residue (DC-Hep) were prepared [25] and examined for their heparanase-inhibiting activity and effect on tumor growth and metastasis in mouse models. After screening a series of DC-Hep compounds with regard to the degree of deacetylation and carboxylation, molecular size, and total charge (beyond the scope of this manuscript), the lead compound (XII) was selected for the studies presented herein. The new compound had a similar safety profile as Roneparstat regarding coagulation parameters, yet showed better pharmacokinetics. It is of interest to elucidate its preferential accumulation in the bone/joint. Albeit to a relatively lower level, the compound reached breast tumor tissue, maintaining the same level up to 72 h. Pharmacological studies showed that our newly generated lead compound XII (Figure 1) was well tolerated and highly effective in several cancer models, including models in which Roneparstat exhibited only a small inhibitory effect (i.e., mesothelioma, pancreatic carcinoma). The superiority of compound XII is attributed in part to its increased negative charge, molecular flexibility, improved affinity toward heparanase, and better pharmacokinetics contributed by the carboxylated groups.

Pancreatic ductal adenocarcinoma (PDAC) is the most prevalent form of pancreatic cancer, comprising 85% of cases, is very difficult to treat, and has a 5-year survival rate of less than 5% [45]. PDAC is characterized by rapid tumor growth, late presentation, early metastasis, and significant resistance to conventional treatments [45]. Surgical resection provides a potential cure; however, 80% of PDAC cases are unresectable at the time of diagnosis, and the majority of resected patients succumb to recurrent disease [45]. Despite negative margins after potentially curative resection, patients who express high levels of heparanase in their resected tumors have worse postoperative survival, suggesting that heparanase expression confers a more aggressive tumor phenotype [35]. The efficacy of compound XII was evaluated against a range of PDAC cell lines, including xenograft (immunocompromised mice), syngeneic models, and a PDX model. Notably, in the Mia PaCa-2 and BxPC-3 models, treatment with compound XII was either as effective or even more effective than that with gemcitabine alone (Figure 2A). Unlike the favorable effect of compound XII in the human PA1233 PDX model, compound XII was not effective in the PA1644 model. Notably, PA1233 is characterized by P53R282W mutation and KRASG12D mutation, while PA1644 is P53 wild-type and KRAS wild-type, suggesting that the anti-tumor effect of compound XII may be related to the P53 and KRAS characteristics of the tumors.

In women, breast cancer (BC) is the most commonly diagnosed cancer and the leading cause of cancer death [46]. While major advances have been made toward breast cancer prevention and treatment, the incidence of breast cancer is still increasing globally, encouraging the development of new technologies for early and more accurate diagnosis. Women under the age of 45 account for 11% of all BC diagnoses in the United States and 9% in the United Kingdom [47]. Moreover, a 16% increase in the incidence of BC in women aged 25–49 years is noted since the 1990s [47]. Compound XII was highly effective in models of breast cancer, both syngeneic and xenograft. It showed activity in the MDA-MB-231 triple negative model, effectively inhibiting the growth of s.c. implanted tumors versus untreated controls (50% inhibition at 40 mg/kg, QD) (Figure 3A). Demonstrating efficacy in a triple negative model has positive implications for the clinical use of compound XII given that this population of breast cancer patients is in need of treatment options because they do not respond to HER2 inhibitors or hormonal treatments [48]. In addition to the triple negative xenograft model, compound XII was highly effective in the 4T1 syngeneic orthotopic model. Notably, despite a limited effect on the primary tumor, compound XII exerted a profound, nearly complete inhibition of spontaneous metastasis following resection (mastectomy) of the primary 4T1 orthotopic breast tumor (Figure 3C). Importantly, compound XII significantly increased the survival of mice compared to the untreated group (Figure 3B). This is important given that metastases, appearing at various times after surgical removal of the primary tumor, are the main cause of failure in the treatment of breast cancer. Due to variability and extended periods, companies focus on drugs that inhibit the growth of the primary tumor and tend to avoid clinical trials aimed at combating metastasis. The anti-metastatic effect of compound XII favors treatment during and right after surgical removal of the primary tumor.

Malignant pleural mesothelioma (MPM), the most common form of mesothelioma, is a highly aggressive tumor characterized by diffuse local growth in the thoracic cavity [49]. It has a poor prognosis because of difficulties in early diagnosis and resistance to conventional therapies [50,51]. Heparanase appears to orchestrate tumor microenvironment crosstalk that drives mesothelioma tumor progression and poor patient outcomes. Impressive results were obtained in a model of human mesothelioma where compound XII was effective as a single agent and nearly as potent as cisplatin (Figure 4). This result is very significant and encouraging given the lack of effective treatments against this rare disease. We have previously demonstrated a profound effect of PG545 in mesothelioma models, further supporting the involvement of heparanase in the pathogenesis of mesothelioma [14]. PG545 (= Pixatimod = cholestanol-sulfo-tetrasaccharide) has been demonstrated to possess potent anti-tumor activity in multiple preclinical models of cancer [41]. Mice treated with PG545 experienced body weight loss, thus limiting the maximal tolerated dose to around 20 mg/kg/week, much lower than the dose of compound XII (160 mg/kg/day) applied in the present study. Interestingly, unlike its potent attenuation of myeloma and sarcoma tumor growth [21], Roneparstat (100% N-desulphated, N-reacetylated, and 25% glycol-split heparin) had little or no effect in the mesothelioma model and was less effective than compound XII in the PDAC and breast carcinoma models. Ongoing mode-of-action studies are aimed at elucidating the reason for these differences in potency.

We previously demonstrated that secretion of latent heparanase followed by its cellular re-uptake is a prerequisite for the delivery of latent heparanase to late endosomes and lysosomes and its subsequent proteolytic processing and activation by lysosomal cathepsin L. This consistent behavior indicates that heparanase is secreted but does not normally accumulate extracellularly unless cell membrane HS is removed or competes with heparin/heparin-like compounds [30]. Our results indicate that compound XII not only inhibits heparanase enzymatic activity (Figure 1) but also decreases the intratumoral levels of heparanase protein (Figure 6A) and inhibits the uptake and activation of the latent enzyme (Figure 6B), thus providing additional mechanistic aspects for its potent anti-tumorigenic mode of action. Given the likely ability of compound XII to interact with heparin/HS-binding proteins other than heparanase (i.e., FGF, VEGF, HGF, HB-EGF), one cannot ascribe the observed anti-tumorigenic effects of compound XII solely to its heparanase-inhibiting capacity.

## 5. Conclusions

Our present results provide a strong rationale for a phase I clinical trial aimed at testing the safety and maximal tolerated dose of compound XII in cancer (i.e., mesothelioma, myeloma, breast carcinoma, pancreatic carcinoma).

## 6. Limitations

Given that isogenic mice have limited genetic diversity, they do not represent the genetic heterogeneity found in human cancers. Moreover, the tumor microenvironment, immune system, and metabolism in mice differ significantly from those in humans, together impacting how a tumor responds to a given drug.

## Data Availability

Data are contained within the article.

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
