# Peer review of "A New Synthesized Dicarboxylated Oxy-Heparin Efficiently Attenuates Tumor Growth and Metastasis"

_cells, 2024, doi:10.3390/cells13030211_

Round 1

Reviewer 1 Report

Comments and Suggestions for Authors

The work presented in the cells-2775858 manuscript describes the effect of a modified heparin derivative in different models of cancer in vivo. The results indicate a significant therapeutic effect of the heparin mimetic in all models tested, showing a clear attenuation of tumor growth and metastasis effect. The manuscript is well-written and the conclusions are supported by the results. 

I just would like the authors to provide a brief comment on the use of models of cancer using isogenic mice and comment on the following points: 

significant limitations can be considered when using isogenic mice to model human cancer treatments:

  1. Lack of genetic diversity: Humans are genetically diverse, and this diversity can significantly influence disease progression and response to treatment. Isogenic mouse models do not capture this variability, which can be a critical factor in the efficacy and toxicity of antitumor drugs.

  2. Tumor heterogeneity: Human tumors are often genetically diverse within the same host, which can affect drug response. Isogenic mice may not adequately model this intra-tumor heterogeneity.

  3. Differences in metabolism: Mice and humans can metabolize drugs differently. Even within the same strain, mice may not accurately reflect the pharmacokinetics and pharmacodynamics of drugs in humans.

  4. Immune response: The immune system's role in cancer and its response to therapy is complex and can differ significantly between mice and humans. The C57BL/6 strain has certain immune characteristics that may not be representative of human immune responses.

  1. I believe the work is interesting and deserves to be published. But I would greatly appreciate to learn the comments of the authors on the points raised above.

Reviewer 2 Report

Comments and Suggestions for Authors

The cloning of the human heparanase gene was published in 1999 in Nature Medicine in two consecutive articles (Vlodavsky et al and Hulett et al., Nature medicine 5, 1999, 799-802 and 803-809 respectively). These articles already highlighted the relevance of the molecule in tumor progression and metastasis and, since then, considerable efforts have been devoted to the development of inhibitors of the molecule that may be of clinical interest. Following this line of research, in this article the authors describe the effect of the compound Oxy-Dicarboxylated Heparin. The authors synthesize the compound, observe its inhibitory effect on the enzymatic activity of heparanase through various procedures, as well as its ability to interfere with the absorption and processing of the molecule by cervical carcinoma cells. Finally, in animal models they examine the effect of the compound on various types of tumors, as well as its pharmacokinetics.

The review of this article leads to raising certain questions related to formal aspects of the manuscript, and others related to the data described.

The manuscript presents four differentiated blocks of information, the first of which are in vitro studies of enzyme inhibition (including the synthesis of the inhibitor), the second its effect in animal models on different tumors, the third the study in a cellular model of the effect of inhibition of uptake and processing of latent heparanase, and the fourth the analysis in an animal model of the effect on coagulation parameters and pharmacokinetics. I find this arrangement of blocks not very orderly, since I think it would be more logical to start with molecular studies, move on to cellular studies and end with the animal model.

Comments related to “materials and methods”:

This part of the manuscript begins with a section titled “2.1. heparanase enzymatic activity” in which the enzyme testing methods are described, but which includes a subsection called “2.1.4. cells” in which the cell types that will be used in tumor models and other studies not related to enzymatic activity are described. The sections should be restructured appropriately.

Something analogous occurs in section 2.2, in which the “animal studies” are described after the description of the animal tumor models. Here too a logical order should be followed.

Throughout the article, the use of recombinant heparanase is mentioned, both in its active and latent forms. Despite its relevance, no section is dedicated to describing the mode of production of the molecule. In line 101 it is mentioned that it was produced “as described (35)”, in line 105 that “was purified and used for the assay”, and in line 472 that “recombinant latent heparanase was added”. It would be important to clearly explain in a section of the methods the procedure for obtaining and purifying both heparanase molecules, the latent and the catalytically active one or, at least, cite in detail the references of both, highlighting the possible modifications of the method used.

The different heparanase activity assay methods described use different pHs: 5.0 (Fondaparinux), 6.0 (Sulfated-labeled ECM), and 7.4 (FRET analysis). The use of differences of up to almost 2.5 pH units is striking, especially when previous studies place the optimal pH of the enzyme between 4.2 (Toyoshima M, Nakajima M. J Biol Chem. 1999, 274:24153-60 ) and 5.1 (Huang KS, et al. Anal Biochem. 2004;333:389-98). These data make sense when considering that heparanase activity develops in acidic environments (lysosomes: pH 4.7-4.8; tumor interior: 5.6-6.8). What is the reason for this large variation? In particular, the tests at a pH as alkaline as 7.4 draw attention, since previous studies describe very small activities at these values.

The concentration of heparanase used in the assays is sometimes described in terms of molarity (Fondaparinux), sometimes in ng/ml (Sulfated-labeled ECM), and sometimes is not made explicit (FRET analysis). A homogeneous description should be used in all cases.

On line 131 a column of Sepharose 6B is cited, but its characteristics are not specified. They should be clearly described.

Verb tenses are not homogeneous. Some paragraphs are written in the present tense, while others are written in the past tense. This should be reviewed and homogenized

Analysis using PMOD is mentioned on line 240, and is mentioned again on line 546. However, at no point is this software described, nor is its manufacturer or its accessibility. This should be mentioned in the text.

Comments related to “results”:

Section 3.1 aims to describe the preparation procedure of the compound studied, but it is only described superficially, although reference 33 is mentioned. The methodology used should be defined in the methods section and, in results, the characterization of the structure, concentration and purity of the molecule obtained

In the inhibition of ECM degradation described in 3.2 and figure 1C, the authors state that peak II is constituted by fractions 20-35. They do not describe, nor is it evident by looking at Figure 1C, what they consider to be peak I, y sería interesante para el lector esa información (solo aparece una mención en la metodología de las referencias 38 y 39). Furthermore, given that it is about solubilizing elements of the ECM through enzymatic fragmentation, should not the set of soluble fractions be considered? In fact, it is striking that Figure 1 mentions that the HS degradation fragments are fractions 17-33, which does not coincide with the main text.

The color code used in Figures 2C and 2D deceives the reader. In Figure 2C, red is used for SST and green is used for XII. However, in 2D, red is used for lower concentrations, and green for higher ones (although it is not specified in the indicative text). It would be better to use homogeneous colors in both figures.

Figure 1 indicates that the ECM degradation assay was carried out for 4 h and 2 ng/ml of recombinant heparanase was used. This does not seem to coincide with what is described in the methods, where incubation times of 3-5 hours and enzyme concentrations of 0.5 ng/ml are described.

In the results sections in which the effects of the compound in tumor models are described, there are paragraphs whose content is not appropriate for this section, but is more appropriate for other sections of the article such as, depending on each case, the introduction or discussion. . E.g. This is the case of lines 290-298 and 330-334 of section 3.3, 348-354 of 3.4, or 405-412 of 3.6.

In lines 358-363 a description of the procedure followed with the MDA-MB-231 human breast carcinoma cells is carried out. This description should be included in the methods section, where, however, the brief mention made (lines 177-175) does not seem to coincide. This also occurs in the description of the methodology referred to 4Q1 included in results, which in part has already been described in methods, and in part is different. The information contained in lines 372-376 should also be included in methods.

The text of lines 440-444 is identical to that already described in lines 188-191.

In section “3.3. pancreatic cancer”, 4 cell lines are mentioned, 3 of human origin (Mia PaCa-2, BxPC-3, AsPC-1) and one of mouse (Panc02). Panc02 is referenced in section “2.1.4. Cells”, but not in “2.2.1. PDAC”, where the other 3 are mentioned. Despite the protocol described in this last section, common to the 3 human cell lines, the results only show the data of PaCa-2, BxPC-3. With respect to AsPC-1 and Panc02, the “median survival prolongation” is mentioned, and the number is simply referred to but the results are not shown. Furthermore, the concentrations of gemcitabine and XII used are different for PaCa-2, BxPC-3. The authors should explain these discrepancies and correct potential errors. Again, the methodology should be clearly stated in the methods section, and not in results.

In relation to figure 4, where the mesothelioma results appear, the way of representing the results is changed, changing the line diagrams for bar diagrams corresponding to days 10 and 23. The results for  cis are not shown in the day 10  figure.  In the photos of mice from day 10, 2 cohorts treated with PBS and another two with XII appear, when the text indicates “mice were divided into 3 groups (n= 6-8/group)”. The authors should review these data.

The deviations do not appear in Figure 5.

In section 3.8. The authors detect heparanase levels in tumors using western blot. However, they do not describe the technique in methods, and the description of the antibodies used is reduced to indicating that they were “polyclonal antibodies”. On the other hand, in the presentation of results in Figure 6, it is striking that the lane on the left, indicated as HPA-22040 8, shows a band located at a different level from those that appear in groups 01-04. Furthermore, none of the bands is located at the level of 50 kDa or 65 kDa, as has been seen in previous studies (e.g. Zetser A. et al. J Cell Sci. 2004;117:2249-58). In the same figure, a statistical analysis of the bars of the histogram on the right does not appear. Are there significant statistical differences between the different concentrations of XII? If so, to what would the authors attribute these differences?

Regarding the inhibition of heparanase uptake and processing, shown in Figure 6B, was only the 24 hour time tested? It would be convenient to see lower times, as well as detect the 65 kDa form in the supernatant.

Regarding pharmacokinetics experiments, why is a lower concentration of compound used than that used in the previous studies with tumors?

Comments related to “discusión

In lines 576-581 it is indicated that “dicarboxylated oxy heparins containing three carboxylate groups per split residue (DC-Hep) were prepared and examined for their heparanase-inhibiting activity. After screening a series of DC-Hep compounds with regard to the degree of deacetylation and carboxylation, molecular size and total charge, the lead compound (XII) was selected for the studies presented.” However, nothing about those screenings is explained anywhere in the text. It would be important to specify them in more detail.

On the other hand, some abbreviations that appear in the text should be defined to facilitate reading, such as, for example, PTT (line 495) or PET/CT (line 527).

Reviewer 3 Report

Comments and Suggestions for Authors The current manuscript entitled “A new synthesized dicarboxylated oxy-heparin efficiently attenuates tumor growth and metastasis (Li et al.)” demonstrates outstanding anti-tumor effect of a new compound XII (DCoxHs), i.e., inhibitory effect of tumor growth and spontaneous metastasis on several cell lines (PDCA (Mia PaCa-2, BxPC-3, PA1233 (PDX model)), breast cancers (MDA-MB-231, 4T1), mesothelioma (MSTO-211H), and myeloma (MPC-11)). Carbohydrate-based anti-tumor therapy, a novel approach in this field, will attract much attention to readers.

Comments.
1. Information on molecular nature of the compound XII is lacking in the text. The reference 33 partially describes molecular origin of the dicarboxylated oxy-heparin (Table 20.2 of ref 33). There seem to be six candidates corresponding to the compound XII. How was the molecular weight, sulfation ratio, and carboxylation ratio of the molecule? These points should be analytically checked and stated in the text or in a supplementary information. Table 20.2  indicates the %DC/UA, therefore it is likely that some analytical identification has been already conducted. This point should be included in the present study.
2. The reference 33 also suggests that the anti-heparanase activity was relatively lower than Roneparstat. However, the anti-tumor effects indicated in the study were comparable to, or better than, Roneparstat. It might be due, for example, to its better pharmacokinetics, but data or reference on Roneparstat is lacking, which makes it difficult to comparatively discuss on the point.   3. It is not likely that the target for the compound XII is restricted to heparanase but other heparin-binding effector molecules and adhesion molecules. Heparanase is involved in the in vivo experimental systems used in the study? Heparanase expression was examined in Mia PaCa-2 (Figure 6), in which the expression level of heparanase varied in each individual. How was the expression of heparanase in other cell lines used in the study, i.e., BxPC-3, MDA-MB-231, 4T1, MSTO-211H, and MPC-11?    A minor comment. 4. Figure 6 panel B lacks experimental condition with heparanase (1 µg/ml) without (0 µg/ml) XII.

Round 2

Reviewer 2 Report

Comments and Suggestions for Authors

After all the modifications made by the authors, many of the important deficiencies and errors in the manuscript have been corrected. However, this reviewer still has some questions to raise.

First of all, I wish to regret the inaccuracies contained in the authors' responses. In all cases, when they refer to the line numbers of the corrections made, these do not coincide with the revised text. Sometimes they even simply don't appear.

In response to the question about the different pHs used in the different enzyme activity testing methods used in the text, the authors state that “Sorry, but we made a mistake in describing the FRET assay. While the test compounds were dissolved and diluted in Tris-HCl (pH 7.4), the final pH of the reaction was 5.5 in a final volume of 12 ul (lines 107-108), as described in ref. #28 (new ref)”. But on lines 107-108, the text describes the Fondaparinux test method, not the FRET. I imagine that the authors are referring to lines 115-116, where it is mentioned. On the other hand, the test conditions are not described in the new reference 28. Furthermore, this reference is not mentioned in relation to the FRET assay, but rather in relation to the heparanase purification method (line 222).

In relation to the question about the analysis using PMOD, the authors specify “The information regarding the PMOD software is now inserted in the text (line 215 and 230 in the revised version.” However, those lines describe the western blot analysis I think the authors are referring to line 247.

In relation to the preparation procedure of the compound studied, the authors indicate “We have now added a section describing the preparation of compound XII (Materials and Methods, lines 82-89). More information is presented in the 'Results' section (lines 234-239)”. Actually, the preparation of the compound appears on lines 88-96 (new), as well as 256-261 (identical to the previous version). The authors also indicate that “Detailed information about the synthesis and chemical properties of the compound is beyond the scope of the current study (now stated on line 523) and may be published at a later time when the patent issue is settled.” None of that is indicated on line 523. On the other hand, the description given in materials and methods of the preparation of the compound is very generic, and does not explain the procedure in detail as one would expect from a scientific text. If this imprecision is due to the authors wishing to patent the method, it would be logical for them to first prepare the patent and then refer to it in the text of the subsequent article.

Confusion for the evaluator also occurs in the answers to other questions, where the line numbers cited by the authors do not coincide with the revised text, forcing a search for each specific aspect throughout the text. This makes the review work difficult, and shows haste in preparing the answers.
